# Effects of Drought Stress on Abscisic Acid Content and Its Related Transcripts in *Allium fistulosum*—*A. cepa* Monosomic Addition Lines

**DOI:** 10.3390/genes15060754

**Published:** 2024-06-08

**Authors:** Tetsuya Nakajima, Shigenori Yaguchi, Sho Hirata, Mostafa Abdelrahman, Tomomi Wada, Ryosuke Mega, Masayoshi Shigyo

**Affiliations:** 1Laboratory of Vegetable Crop Science, Division of Life Science, Graduate School of Sciences and Technology for Innovation, Yamaguchi University, Yamaguchi 753-8515, Japan; c004wfw@yamaguchi-u.ac.jp; 2Department of Food Science and Technology, National Fisheries University, 2-7-1 Nagata-Honmachi, Shimonoseki 759-6595, Japan; yaguchi@fish-u.ac.jp; 3Laboratory of Agroecology, Department of Bioresource Sciences, Faculty of Agriculture, Kyushu University, 744 Motooka, Nishi-ku, Fukuoka 819-0395, Japan; hirata.sho.481@m.kyushu-u.ac.jp; 4Institute of Genomics for Crop Abiotic Stress Tolerance, Department of Plant and Soil Science, Texas Tech University, Lubbock, TX 79409, USA; mosabdel@ttu.edu; 5Laboratory of Plant Breeding, Faculty of Agriculture Yamaguchi University, Yamaguchi 753-8515, Japan; wadat538@affrc.go.jp (T.W.); mega@yamaguchi-u.ac.jp (R.M.); 6Laboratory of Plant Breeding, Division of Life Science, Graduate School of Sciences and Technology for Innovation, Yamaguchi University, Yamaguchi 753-8515, Japan

**Keywords:** Japanese bunching onion, alien monosomic addition, drought stress, ABA, *β*-carotene

## Abstract

Climate change has resulted in an increased demand for Japanese bunching onions *(Allium fistulosum* L., genomes FF) with drought resistance. A complete set of alien monosomic addition lines of *A. fistulosum* with extra chromosomes from shallot (*A. cepa* L. Aggregatum group, AA), represented as FF + 1A–FF + 8A, displays a variety of phenotypes that significantly differ from those of the recipient species. In this study, we investigated the impact of drought stress on abscisic acid (ABA) and its precursor, β-carotene, utilizing this complete set. In addition, we analyzed the expression levels of genes related to ABA biosynthesis, catabolism, and drought stress signal transduction in FF + 1A and FF + 6A, which show characteristic variations in ABA accumulation. A number of unigenes related to ABA were selected through a database using *Allium* TDB. Under drought conditions, FF + 1A exhibited significantly higher ABA and β-carotene content compared with FF. Additionally, the expression levels of all ABA-related genes in FF + 1A were higher than those in FF. These results indicate that the addition of chromosome 1A from shallot caused the high expression of ABA biosynthesis genes, leading to increased levels of ABA accumulation. Therefore, it is expected that the introduction of alien genes from the shallot will upwardly modify ABA content, which is directly related to stomatal closure, leading to drought stress tolerance in FF.

## 1. Introduction

The Japanese bunching onion (*A. fistulosum* L., FF), also named the Welsh onion, spring onion, and scallion, is widely distributed from Siberia to tropical Asia, especially in east Asia, with many varieties adapted to local environmental conditions [1,2,3]. FF is relatively tolerant to heat, cold, and drought conditions, but it is susceptible to waterlogging [4]. However, in recent summer production, high levels of light and drought have retarded growth, resulting in lower yields [5]. Additionally, the process of withholding irrigation just before harvest has caused the wilting of leaf tips, resulting in the loss of product value. With global warming and desertification becoming problems due to climate variation, drought-resistant traits are required for the stable production of FF. Plants have unique ways of responding to various external environmental stresses; among them, the plant hormone abscisic acid (ABA) plays a role in plant dormancy and drought response [6]. Studies on the drought response of ABA have been conducted in a variety of plants. For example, exogenous ABA treatment can regulate the processes of metabolizing energy, amino acids, and lipids; promote the accumulation of flavonoids, betaine, and other substances; improve enzymatic and non-enzymatic antioxidant regulation systems; and enhance the photosynthetic performance and relative water content of plants to promote plant growth and improve drought resistance of diverse crops, including maize (*Zea mays*), wheat (*Triticum aestivum*), sweet potatoes (*Ipomoea batatas*), and pearl millet (*Pennisetum glaucum*) [7,8,9,10]. Additionally, plants exposed to abiotic stress swiftly activate the ABA signaling cascade, leading to the activation of ABA-responsive transcription factors and genes, promoting stomatal closure, modifying root architecture, and affecting the expression of stress-responsive genes and physiological responses [11]. Once stress tolerance is achieved, it is essential to terminate or attenuate the ABA pathway. This balance is maintained through the selective degradation of specific components of the ABA signaling pathway, marked for decay by ubiquitination [12]. Furthermore, an additional layer of complexity is introduced by the extensive crosstalk between ABA and other phytohormones, such as cytokinins, ethylene, jasmonic acid, and gibberellins, to maintain the balance between stress adaption and growth [13]. These mechanisms collectively underscore the importance of ABA in enhancing plant resilience to drought stress. By elucidating the role of ABA and its associated pathways, we aim to contribute to the development of drought-tolerant crops, which is vital for agricultural sustainability in the face of climate change [14]. The shallot is a species of subtropical origin that has been recognized as a potential genetic resource for improving *Allium* crops because of its adaptability to environmental stresses [15,16]. Alien monosomic addition lines (AMALs) are valuable for elucidating genome structure and transferring useful genes and traits in plant breeding. AMALs were used in breeding programs of rice (*Oryza sativa* L.), oilseed rape (*Brassica napus* L.), and wheat [17,18,19,20]. Shigyo et al. [21] have generated a series of AMALs in FF strains by introducing shallot chromosomes. These lines have been investigated to study the morphological features, primary metabolites, secondary metabolites, and other biochemical characteristics influenced by each shallot chromosome addition. Furthermore, the bunching onion line FF + 1A, with an added shallot, has shown high resistance to rust disease, while FF + 2A exhibits resistance to onion yellow dwarf virus, as has been reported [22,23]. The impacts of AMALs on the biotic factor have become evident. Therefore, this study aims to investigate the impact of drought stress on ABA and its precursor, β-carotene, using a complete set of AMALs of *A. fistulosum* with extra chromosomes from shallot. We analyze the expression levels of genes related to ABA biosynthesis, catabolism, and drought stress signal transduction in AMALs, particularly FF + 1A and FF + 6A, which show characteristic variation in ABA accumulation. By examining both ABA and β-carotene, we aim to provide a comprehensive understanding of the biosynthetic pathway and the role of ABA in drought stress tolerance in FF.

## 2. Materials and Methods

### 2.1. Plant Materials and Drought Treatment

The plant materials—eight different AMALs (2*n* = 2*x* + 1 = 17; FF + 1A, FF + 2A, FF + 3A, FF + 4A, FF + 5A, FF + 6A, FF + 7A, FF + 8A) [21] and *A. fistulosum* (FF)—were grown in 6-inch clay pots filled with sand. Each pot contained one plant, and all of the plant genotypes were vegetatively propagated in the greenhouse of Yamaguchi University (34° N, 131° E). Pots were randomly arranged on a rack, watered every two days, and fertilized with 1000× HYPONeX solution (Hyponex Japan, Osaka, Japan) once a week until the drought test. Watering was accomplished by slowly filling the pot until the water overflowed the top (approximately 300 mL/pot). In January 2019, the drought treatment was implemented by stopping irrigation for 30 days, while the control condition was implemented by irrigating every 2 days. Plants were collected from each line separately in biological replicates (*n* = 5) for each condition. Leaf tips, excluding the yellowed parts, were collected at approximately 10 cm and were frozen in liquid nitrogen. The frozen samples were powdered using a mortar and pestle; samples were then dried using a freeze dryer (TAITEC VD-250R freeze dryer coupled with a vacuum pump, TAITEC, Saitama, Japan). These dried samples were used for measuring β-carotene, violaxanthin, neoxanthin, and ABA, as well as quantitative qRT-PCR (qPCR) of genes involved in ABA biosynthesis and catabolism and drought stress signal transduction.

### 2.2. β-Carotene, Violaxanthin, and Neoxanthin Measurement

Briefly, 4 mg of dried sample was accurately weighed and transferred to a 1.5 mL tube, to which 500 μL of cold 100% acetone was added. The solution was vortexed for 2 min and ultrasonicated for 20 min in a cool condition. The extract was centrifuged at 5000 rpm for 5 min at 10 °C, and the supernatant was collected. Again, 100% cold acetone was added to the remaining residue of the lower layer, and the same operation was performed twice to obtain the supernatant. All supernatants obtained were filtered through filter tubes (Nanosep^®^ centrifugal devices, Pall Corportaion, NewYork, NY, USA) and used as sample solutions for HPLC. The sample solution was stored at −20 °C in a freezer and used for measurement within 3 days. The extracts were analyzed by HPLC equipped with a UV–Vis detector (HITACHI L7420, Hitachi, Tokyo, Japan) operating at 435 nm to detect the carotenoids. The carotenoids were separated on LiChroCART 250-4.0 Lichrospher 100RP-18, 5 μm (KANTO CHEMICAL, Tokyo, Japan). The separation was achieved by gradient elution with (A) 80% MeOH solution (mixing 400 mL of methanol (HPLC-grade)), 50 mL of ultrapure water, and 50 mL of 100 μM HEPES (pH 7.5) and (B) ethyl acetate. The modified gradient elution program was run as follows: (i) the initial conditions were 100 % (A), (ii) a 20 min linear gradient to 50% (A) and 50% (B), and (iii) 50% (A) and 50% (B) for 30 min, and the flow rate was 1 mL/minute. The column temperature was set to 30 °C, and the injection volume was 20 μL.

### 2.3. ABA Measurement

Briefly, 2 mg of each dried sample was accurately weighed and transferred to a 1.5 mL tube, to which 250 μL of 80% methanol and 25 μL of 1 ppm d6ABA (Toronto Research Chemicals, Toronto, ON, Canada) were added as an internal standard. The solution was vortexed and incubated overnight at room temperature under light protection. The extract was then centrifuged at 15,000 rpm for 5 min at 4 °C, and the supernatant was filtered through filter tubes (Nanosep^®^ centrifugal devices, Pall Corporation, NewYork, NY, USA) and used as a sample solution for LC-MS/MS. The samples were analyzed by a Prominence Modular HPLC (SHIMADZU, Kyoto, Japan) equipped with Mightysil RP-18 GP (II) 150-2.0, 5 μm (KANTO CHEMICAL, Tokyo, Japan), eluted with a 0.2 mL/min binary gradient containing the mobile phases (A) 0.1% formic acid solution (LCMS-grade) dissolved in ultrapure water and (B) 0.1% formic acid solution (LCMS-grade) dissolved in acetonitrile (LCMS-grade). The modified gradient elution program was performed as follows: (i) the initial conditions were 80% (A) and 20% (B) for 5 min, (ii) a 15 min linear gradient to 10% (A) and 90% (B), (iii) 10 % (A) and 90% (B) for 10 min, and (iv) a 5 min linear gradient to 80% (A) and 20%. The column temperature was set at 40 °C, and the injection volume was 4 μL. Mass spectrometric analysis was carried out on a triple-quadrupole 3200 QTrap mass spectrometer (SCIEX, Framingham, MA, USA) equipped with negative electrospray ionization (ESI).

### 2.4. qPCR

Total RNA was extracted using an RNeasy Plant Mini Kit (QIAGEN Sciences, Hilden, Germany). RNA quality was assessed using a NanoDrop (Thermo Fisher, Waltham, MA, USA). The cDNA library was constructed using ReverTra Ace qPCR RT Master Mix with gDNA Remover (TOYOBO, Osaka, Japan) in accordance with the manufacturer’s instructions. qPCR was used to measure transcript abundance for *β-carotene hydroxylase1* (*BCH1*), *zeaxanthin epoxidase* (*ABA1*), *nine-cis-epoxycarotenoid dioxygenase 3* (*NCED3*), *xanthoxin dehydrogenase* (*ABA2*), *abscisic-aldehyde oxidase* (*AAO3*), and *molybdenum cofactor sulfurase ABA3* as ABA biosynthesis genes, *ABA 8’-hydroxylase* (*CYP707A1*, *CYP707A3*) as ABA catabolism genes, *transducin*, and *late embryogenesis abundant* (*LEA4-5*) genes as drought stress signal transduction genes in drought (Figure 1) conditions and control conditions. qPCR was performed using THUNDERBIRD SYBR qPCR Mix (QPS-201, TOYOBO, Osaka, Japan) on a QuantStudio 1 instrument. Amplification was performed using the following cycling parameters: 1 cycle of 94 °C for 1 s, 40 cycles of 95 °C for 15 s, 60 °C for 60 s, and 95 °C for 1 s. Fluorescence was acquired at 60 °C. The relative transcript levels were calculated using the comparative $C_{t}$ (Threshold Cycle) method, with the β-actin gene as an internal control. Results were obtained by excluding the highest $C_{t}$ value and the lowest $C_{t}$ value and calculating the mean for three replicates.

### 2.5. Primer Design

The genes involved in ABA biosynthesis, catabolism, and signal transduction in the *Allium* genus were investigated using the *Allium* genus transcriptome database known as *Allium* TDB (http://alliumtdb.kazusa.or.jp/index.html accessed on 2 May 2024). Specifically, gene searches were conducted through keyword searches in *Allium* TDB, and Unified Genes from a DHA (doubled-haploid of shallot) bulb [24], with similarity to gzenes in the *Arabidopsis* Information Resource (TAIR) database (https://www.arabidopsis.org/ accessed on 2 May 2024), were obtained. Subsequently, primers for qPCR were designed using the obtained Unified Genes’ information from FF that matched the identified DHA bulb genes (Table 1).

### 2.6. Ascorbic Acid Measurement

The ascorbic acid content was measured monthly from April 2005 to March 2006 using fresh leaf blades of FF, 1A, and 6A as plant materials to obtain a comprehensive and detailed understanding of the seasonal variations in ascorbic acid levels. This year-long assessment allowed us to capture any fluctuations due to environmental changes over the seasons, providing a robust dataset for analysis. The ascorbic acid content was detected using a previously described method [25].

### 2.7. Statistical Analysis

Statistical analyses were performed with *SPSS* Ver28. Each sample was measured once, and the data are expressed as the mean ± the standard error (SE). Calculations were performed using one-way ANOVA and Dunnett’s multiple comparison test or Student’s *t*-test. Dunnett’s multiple comparison test was designed for comparing the difference the mean value of each line, and Student’s *t*-test was designed for comparing the difference of the two conditions; *p* < 0.05 was considered as statistically significant.

## 3. Results

### 3.1. Carotenoid and ABA Content in AMALs under Drought Conditions

As a result of the drought treatment, all 45 individuals (FF and AMALs with five replicates each) survived. Measurements of β-carotene under both control and drought conditions revealed significantly higher β-carotene content in FF + 1A compared with FF and other AMALs (Figure 2A; Appendix A). On the other hand, β-carotene content decreased significantly (*p* < 0.05) under drought versus control conditions in all investigated AMALs, except for FF and FF + 5A, which did not show any significant changes (Figure 2A). Similarly, ABA content increased significantly in all investigated AMALs under drought versus control conditions, with FF + 1A exhibiting the highest ABA accumulation (Figure 2B; Appendix A). On the other hand, only the FF + 6A line showed the highest ABA accumulation under control conditions compared to FF and the other AMALs (Figure 2B). Additionally, a moderate positive correlation was observed between β-carotene and ABA contents under drought conditions (Figure 3). Additionally, when comparing the amount of violaxanthin and neoxanthin for FF, FF + 1A, and FF + 6A, no significant differences were observed between the control and drought conditions (Figure 4). However, under drought conditions for FF + 6A, the peaks of violaxanthin and neoxanthin were not detected.

### 3.2. Expression Analysis of ABA-Related Genes in the Monosomic Addition Lines and A. fistulosum under Drought Conditions

Under control conditions, FF + 1A showed an approximately twofold increase in the expression of genes involved in biosynthesis, catabolism, and drought stress signal transduction as compared to FF. In FF + 6A, genes related to biosynthesis (except *NCED3* and *ABA3*) and catabolism showed a decrease, with catabolism-related genes exhibiting a reduction of more than twofold. Genes associated with signaling transduction did not show changes as compared to FF (Figure 5; Appendix A). Conversely, under drought conditions, FF and FF + 6A showed similar trends in gene expression, characterized by a substantial upregulation of genes associated with biosynthesis and a downregulation of genes related to catabolism as compared to the FF control. Particularly noteworthy were the substantial elevations observed in *BCH1* and *NCED3*, while *ABA1* remained unaltered. Regarding signaling, *LEA4-5* displayed an approximately onefold decrease, whereas *transducin* showed an increase.

### 3.3. Determination of Ascorbic Acid Content in FF, FF + 1A, and FF + 6A

The results of Dunnett’s multiple comparison test comparing the annual average levels of ascorbic acid content for FF, FF + 1A, and FF + 6A showed that FF + 1A exhibited significantly higher values as compared to FF (Figure 6; Appendix A).

## 4. Discussion

Our results showed three patterns: (i) the addition of chromosome 1A into the FF genome increased the expression of ABA-related genes under control conditions without changes in ABA levels; (ii) FF + 6A under control conditions decreased gene expression, but there was no accumulation of ABA; (iii) FF + 1A under drought conditions increased both gene expression and ABA accumulation (Figure 2B and Figure 5). It has been argued that ABA levels are maintained by a balance between its biosynthesis and catabolism, rather than solely by biosynthesis [26]. Therefore, these patterns can be explained by the balance of the expression levels of genes for biosynthesis and catabolism. Expression analysis in FF + 1A showed higher expression levels of all genes related to biosynthesis, catabolism, and drought stress signal transduction as compared to FF; in particular, higher expression levels were shown in drought conditions. The genomic sequence information of bunching onions and onions that have similarity to shallots was reported by Liao et al. [27] and Fei Hao et al. [28]. This information revealed that ABA biosynthesis genes *BCH1*, *NCED3*, and *transducin* were located on the first chromosome of the bunching onion (Table 1). On the other hand, genes involved in ABA biosynthesis and catabolism were not located on the first chromosome of the shallot. Therefore, the characteristic expression of genes involved in ABA biosynthesis and catabolism in 1A is likely to be the result of the interaction between the additional first chromosome of the shallot and the chromosomes of *A. fistulosum*. Previous study for the sugar beet monosomic addition line M14, derived from a cross between *Beta vulgaris* L. and *B. coloriflora* Zoss, have reported that the addition f the chromosome affects the gene expression of the recipient chromosome [29]. Additionally, FF + 1A exhibited high expression of the drought stress signal transduction *LEA4-5*, not only in drought conditions, but also in the control condition, where ABA did not accumulate. LEA proteins are known to confer resistance to drought, low temperature, and high salt concentrations; increased expression of *LEA4* has been shown to enhance resistance to drought, salt, and heavy metal stress in rice [30] and Arabidopsis [31]. Therefore, the high expression of *LEA4-5* in FF + 1A during both control and drought conditions suggests the potential for stress resistance to drought and various abiotic stresses.

*ABA1* was highly expressed only in FF + 1A, while no increase was observed in FF and FF + 6A under drought conditions, and ABA accumulation did not progress. Under drought stress, ABA1 increased in the roots, but not in the leaves of *Nicotiana plumbaginifolia* var *Viviani* (*ABA2*/*NpZEP*) and tomato (*Solanum lycopersicum*) (*LeZEP1*) [32,33]. In Arabidopsis, drought stress has been reported to lead to an increase in ZEP protein in roots and the degradation of ZEP protein in leaves [34], which is consistent with the results of FF and FF + 6A in this study. *ABA1* encodes zeaxanthin epoxidase, which converts zeaxanthin to antheraxanthin and violaxanthin, which is related to the xanthophyll cycle, the thermal dissipation system protecting against photoinhibition in plants. Under drought conditions, reduced CO_2_ uptake from stomatal closure decreases dark reaction activity. As a result, the light reaction is higher than the dark reaction, leading to the accumulation of NADPH. Singlet oxygen (^1^O_2_) is generated in photosystem II, while hydrogen peroxide (H_2_O_2_) is generated in photosystem I. Defense mechanisms against such situations include the thermal dissipation system and the scavenging systems of reactive oxygen species. In FF and FF + 6A, the xanthophyll cycle is thought to de-epoxidize, lowering the light-harvesting efficiency and converting excess light energy into heat to prevent photoinhibition. In FF + 1A with a high accumulation of ABA, the scavenging system of reactive oxygen species is likely involved. In plants, antioxidants such as β-carotene and ascorbic acid remove singlet oxygen and hydrogen peroxide, respectively [35,36]. Furthermore, ascorbic acid also plays a role in oxidizing NADPH through the glutathione–ascorbic acid cycle. Ascorbic acid plays a role in increasing stress tolerance in Common bean (*Phaseolus vulgaris* L.), maize, flax (*Linum usitatissimum* L.), wheat, and broccoli (*Brassica oleracea* var. *italica*) [37,38,39,40]. FF + 1A was found to have significantly higher β-carotene content under drought conditions, and it was also confirmed that ascorbic acid tends to be higher throughout the year, indicating a high capacity to remove reactive oxygen species and NADPH, making it less susceptible to photoinhibition. Therefore, the epoxidation of the xanthophyll cycle and ABA biosynthesis likely progressed in FF + 1A under drought conditions. Accordingly, *A. fistulosum* with the extra first chromosome of shallots could be less susceptible to photoinhibition, potentially making it capable of maintaining closed stomata and capable of addressing dryness issues during the summer season. This study highlights the role of ABA in enhancing drought tolerance in *A. fistulosum* through the addition of a shallot chromosome. Our results show that the FF + 1A line exhibits increased ABA-related gene expression and higher ABA accumulation under drought conditions, compared with the FF or FF + 6A line. This suggests a beneficial genetic interaction enhancing the plant drought response mechanisms. Additionally, the high levels of β-carotene and ascorbic acid in FF + 1A under drought conditions indicate a robust reactive oxygen species-scavenging system, reducing photoinhibition and oxidative stress. This adaptation allows the FF + 1A line to maintain efficient photosynthesis during water scarcity. Our findings demonstrate the potential of using AMALs to introduce beneficial traits from related species to improve crop resilience. Specifically, the addition of the first shallot chromosome enhances ABA biosynthesis and stress response pathways, offering a valuable resource for breeding drought-tolerant *A. fistulosum* varieties. In summary, our study underscores the importance of ABA in drought response and the benefits of genetic additions from stress-tolerant species, contributing to more stable crop production amid climate variability. 

## Figures and Tables

**Figure 1 genes-15-00754-f001:**
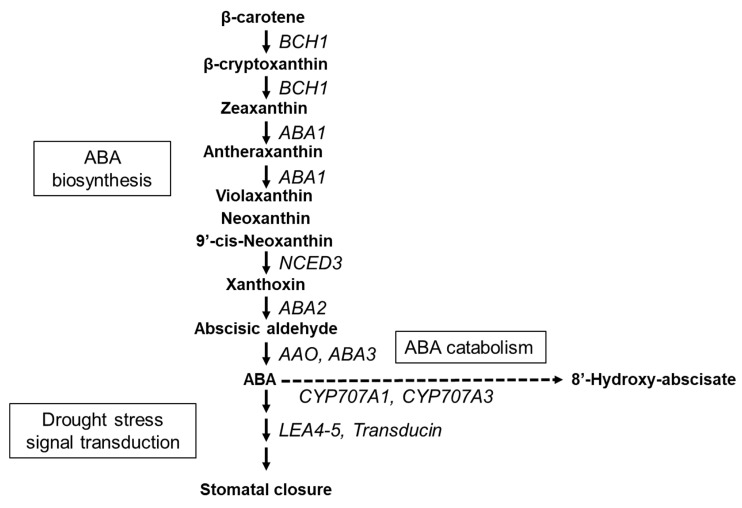
Schematic diagram illustrating the key pathways involved in abscisic acid (ABA) biosynthesis, catabolism, and drought stress signal transduction. *BCH1*: *β-carotene hydroxylase1*; *ABA1*: *zeaxanthin epoxidase*; *NCED3*: *nine-cis-epoxycarotenoid dioxygenase 3*; *ABA2*:*xanthoxin dehydrogenase*; *AAO*: *abscisic-aldehyde oxidase*; *ABA3*: *Molybdenum cofactor sulfurase*; *CYP707A1*: *ABA 8’-hydroxylase1*; *CYP707A3*: *ABA 8’-hydroxylase3*; *LEA4-5*: *late embryogenesis abundant*.

**Figure 2 genes-15-00754-f002:**
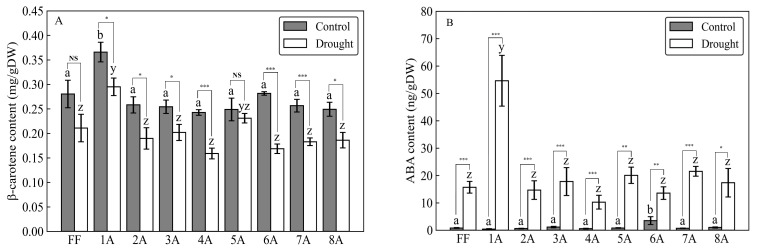
Comparison of (**A**) β-carotene and (**B**) abscisic acid (ABA) contents in different monosomic addition lines (AMALs, 1A, 2A, 3A, 4A, 5A, 6A, 7A, and 8A) and *Allium fistulosum* (FF) under control and drought stress conditions. Different letters (a, b for control conditions and y, z for drought conditions) indicate significant differences at *p* < 0.05. Asterisks *, **, and *** represent significant differences at *p* < 0.05, *p* < 0.01, and *p* < 0.001, respectively, as determined by Student’s *t*-test. NS indicates not significant at *p* < 0.05.

**Figure 3 genes-15-00754-f003:**
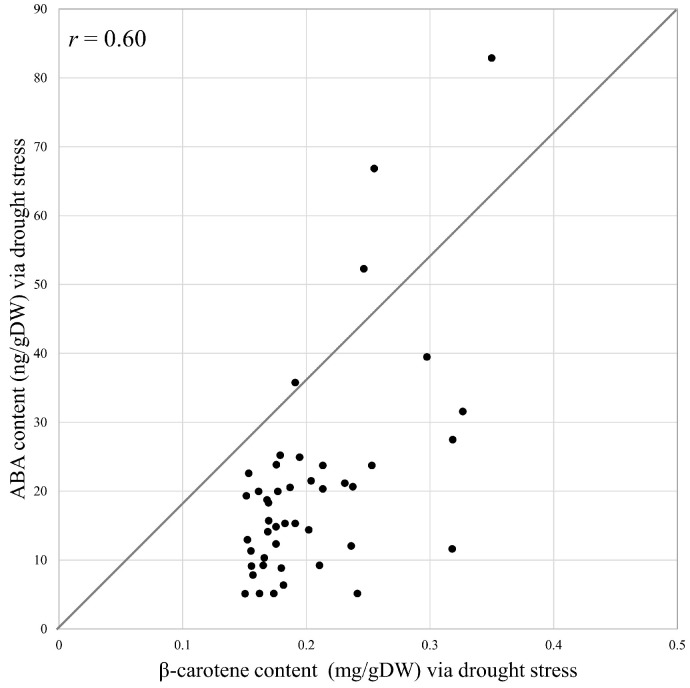
Correlation analysis of β-carotene and abscisic acid (ABA) contents under drought stress in eight different alien monosomic addition lines and *Allium fistulosum* using the Pearson correlation method.

**Figure 4 genes-15-00754-f004:**
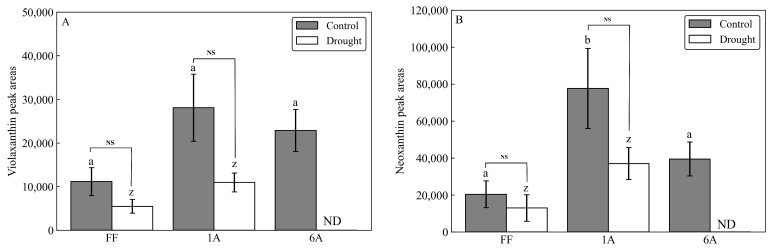
Comparison of (**A**) violaxanthin and (**B**) neoxanthin peak areas in FF, FF + 1A, and FF + 6A under control and drought conditions. Different letters (a, b for control conditions and z for drought conditions) indicate significant differences at *p* < 0.05. NS indicates not significant at *p* < 0.05 by Student’s *t*-test.

**Figure 5 genes-15-00754-f005:**
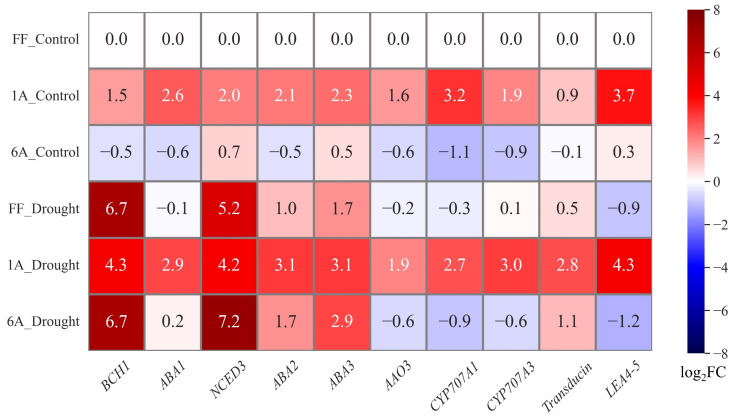
Heatmap of $\log_{2}$ transformed average gene expression changes in abscisic acid (ABA) biosynthesis and catabolism-related genes under control and drought conditions in FF, FF + 1A, and FF + 6A. *BCH1*: *β-carotene hydroxylase1*; *ABA1*: *zeaxanthin epoxidase*; *NCED3*: *nine-cis-epoxycarotenoid dioxygenase 3*; *ABA2*: *xanthoxin dehydrogenase*; *AAO*: *abscisic-aldehyde oxidase*; *ABA3*: *Molybdenum cofactor sulfurase*; *CYP707A1*: *ABA 8’-hydroxylase1*; *CYP707A3*: *ABA 8’-hydroxylase3*; *LEA4-5*: *late embryogenesis abundant.*

**Figure 6 genes-15-00754-f006:**
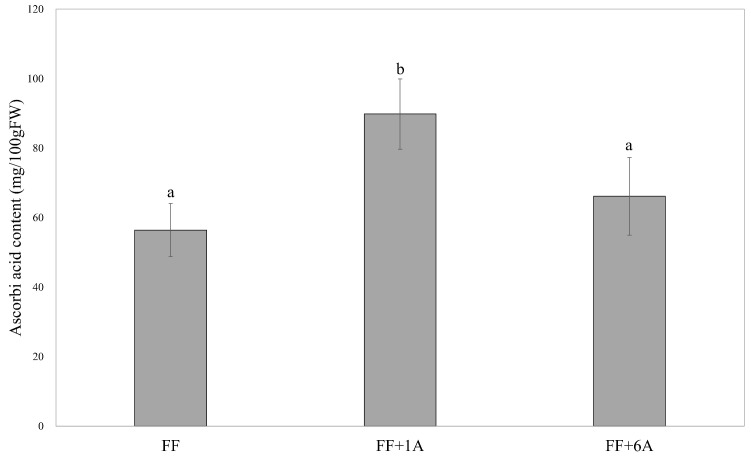
Annual average ascorbic acid content in FF, FF + 1A, and FF + 6A Dunnett’s multiple comparison test was used to compare each line with FF as a control.Values are the means ± SE (*n* = 12). Different small letters (a and b) refer to significant differences (*p* < 0.05).

**Table 1 genes-15-00754-t001:** Primer list for genes used in this study as shown in Figure 1.

Arabidopsis Genome Initiative Code	Gene Name	Homologs						Primer Sequence Based on *A. fistulosum* Unigenes	
		* **A. cepa** * **Unigene** **(bp)**	**Identity to** **Arabidopsis** **Gene (%)**	**Chromosomal** **Location**	* **A. fistulosum** * **Unigene** **(bp)**	**Identity to** * **A. cepa** * **UniGene (%)**	**Chromosomal** **Location**	**Forward (5′ to 3′)**	**Reverse (5′ to 3′)**
AT4G25700.1	BCH1	CL831.Contig1_ DHA_Bulb (1169)	73.4	8A	Unigene22611_ FFStem (339)	97.9	1F, 7F	AGGCAAAAACG AAGCAGCAG	TCGCGGCAACC AAATAAGTG
AT5G67030.2	ABA1	Unigene26642_ DHA_Bulb (687)	82.9	4A, 5A	CL2970.Contig3_ FFStem (1321)	100.0	4F, 5F	TCCTCTTTCTG CAGCAGGTG	AAGAGGTCATG AGTGCTGGC
AT3G14440.1	NCED3	Unigene2060_ DHA_Bulb (237)	59.2	7A	Unigene35308_ FF_5AStem (510)	86.8	1F	GTTGGTTCACC GGTGCAAAG	ACATATTCCAA CAAGCTGCAGC
AT1G52340.1	ABA2	CL2419.Contig1_ DHA_Bulb (799)	62.9	4A	CL736.Contig4_ FFStem (960)	97.7	6F	TGGTGCTCCTG AGACAACAAG	GAGCGTTGATG ACCTCAATAGC
AT2G27150.2	AAO	CL1978.Contig1_ DHA_Bulb (423)	70.1	3A, 5A, 8A	Unigene26780_ FFStem (865)	89.5	2F, 3F, 5F, 7F, 8F	TCCACCAAAAC CTCCTCCAAC	CATGCCAAGCC CCTCAATTTAC
AT1G16540.1	ABA3	Unigene33397_ DHA_Bulb (272)	48.9	4A, 5A, 8A	Unigene34756_ FFStem (260)	97.7	6F	TGGTGCTCCTG AGACAACAAG	GAGCGTTGATG ACCTCAATAGC
AT4G19230.1	CYP707A1	CL2918.Contig1_ DHA_Bulb (1782)	72.0	5A, 8A	CL5541.Contig1_ FFStem (249)	99.2	7F	TTGTGGTCAGG TGATGAAGC	GCACCAAAGCC AAACACTTTC
AT5G45340.2	CYP707A3	Unigene28686_ DHA_Bulb (437)	76.6	4A	Unigene30555_ FFStem (440)	99.3	4F	ACCTTTACCTCC AGGCTCTATG	GGCAACCCAAGA TGTGAGTTTTG
AT1G49450.1	Transducin	Unigene7829_ DHA_Bulb (423)	45.5	1A	Unigene31313_ FFStem (590)	97.8	1F	TGTGTCCACGG CGATTTTAC	AATTGCTCGGTC TGTTCACC
AT1G01470.1	LEA4-5	CL838.Contig2_ DHA_Bulb (610)	53.7	3A, 4A, 6A	Unigene30105_ FFStem (491)	95.7	2F, 5F, 6F, 7F, 8F	TTTCTTGGTCA CTGGAAGCG	TGGTGCCAATG AAAGTGGAC
AT3G53750.1	β-actin	CL575.Contig2_ DHA_Bulb (454)	99.3	-	CL1482.Contig7_ FFStem (351)	98.0	-	GTTGGTATGGG GCAAAAAGA	AGCCTTTGGAT TGAGTGGTG

*BCH1*: *β-carotene hydroxylase1*; *ABA1*: *zeaxanthin epoxidase*; *NCED3*: *nine-cis-epoxycarotenoid dioxygenase 3*; *ABA2*: *xanthoxin dehydrogenase*; *AAO*: *abscisic-aldehyde oxidase*; *ABA3*: *Molybdenum cofactor sulfurase*; *CYP707A1*: *ABA 8’-hydroxylase1*; *CYP707A3*: *ABA 8’-hydroxylase3*; *LEA4-5*: *late embryogenesis abundant*.

## Data Availability

The original contributions presented in this study are included in the article and Appendix A; further inquiries can be directed to the corresponding authors.

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
