# Peer review of "Effects of Drought Stress on Abscisic Acid Content and Its Related Transcripts in Allium fistulosumA. cepa Monosomic Addition Lines"

_genes, 2024, doi:10.3390/genes15060754_

Round 1

Reviewer 1 Report

Comments and Suggestions for Authors

Specific comments:

Lines 26-28: Please add the appropriate reference.

It is necessary to better clarify the aim of the manuscript. Your objective, as stated in your writing, is to investigate how abscisic acid affects MALs under abiotic stress. But you also examined carotenoids; your focus was on the content of ABA during droughts.

The introduction is brief and insufficiently elaborated. If possible, clarify a little better the role of ABA in the response to drought in plants. You mention that ABA regulates various processes and enhances drought tolerance in diverse crops, but please explain how. Why is ABA so important?

Check the Latin names in the paper and put them in italics where they are not (lines 126, 187-189)

The discussion is very informative, however, concrete conclusions based on the obtained data are still lacking. It would be good to state, in a few sentences, at the end the significance of the results obtained in this experiment.

Author Response

Plese see the attachment.

Reviewer 2 Report

Comments and Suggestions for Authors

This article presents interesting results related to the alteration of ABA content and its related transcripts in A. fistulosum - A. cepa monosomic addition lines under water stress.

The adopted approach in this study was very consistent since it could provide more insights into the understanding of JA related genes plant tolerance to low temperature.

The manuscript was well introduced, and the authors adopted convincing methodology with the discussion of the obtained results. However, the manuscript needs substantial revisions.

General comment

Comment 1: The English of this manuscript needs minor editing.

Comment 2: Many details of the methodology are missing.

Other comments

Introduction

- L37: Please correct the citation form.

- The aim of this study is not well highlighted at the end of the Introduction section. Please provide a clear one and add an hypothesis too.

M&M

- L52: “under the same conditions”, please specify these conditions.

- L54: “Drought treatment was imposed by withdrawing irrigation”, It is not clear, did you stop irrigation? If it is true, for how long? Or did you apply a reduced irrigation amount? If so, how much?

-L53-54: You mentioned that you have 5 biological replicates and you stated, “Drought treatment was imposed by withdrawing irrigation for half of the plants”, Do you mean "half of the 5 replicates"?? It is not clear!

- L55: “standard irrigation regime as a control”, what is the “standard irrigation regime”, how much water you applied and how many times?

L67: please add a space between the value and the unit and check throughout the manuscript.

L76: please remove the duplicate bracket.

L116: Please provide the significance of the abbreviation “Ct” at the first appearance and check throughout the manuscript.

L126: Please italicize the species scientific name and check throughout the manuscript.

Table 1: Please provide the significance of the abbreviations as a footnote.

- L128-130: why did you assess ascorbic acid content monthly from April 2005 to March 2006? and why did you choose FF, FF+1A and FF+6A to perform it and not for all lines?

- L132: Please provide more details in Statistical analysis subsection, including the applied significance test.

Results

- The Figures caption should be reviewed to be more precise and clearer.

- Please provide the % of variations in this section to show the variation extent of the measured traits.

- L153-155 and L164-165: the same statement was repeated.

Conclusions

- Please provide a conclusions section with a focus on your results since the last two sentences provided at the end of the discussion section are not enough.

Comments on the Quality of English Language

The English language of this manuscript needs minor editing.

Round 2

Reviewer 2 Report

Comments and Suggestions for Authors

The authors satisfied all the raised comments. I endorse the publication of the current version of the manuscript.